Beliefs in being unlucky and deficits in executive functioning: an ERP study

Martín del Campo Ríos Jaime jm420@le.ac.uk
Fuggetta Giorgio
Maltby John
College of Medicine, Biological Sciences and Psychology, University of Leicester , Leicester , United Kingdom
Yuan Tifei
Electronic publication date: 2015 Jun 25
Publication date: 2015
Volume: 3
Electronic Location ID: e1007
Received 2014 Dec 19; Accepted 2015 May 18
Copyright: © 2015 Martín del Campo et al.
Copyright year: 2015
Copyright holder: Martín del Campo et al.
License: This is an open access article distributed under the terms of the Creative Commons Attribution License, which permits unrestricted use, distribution, reproduction and adaptation in any medium and for any purpose provided that it is properly attributed. For attribution, the original author(s), title, publication source (PeerJ) and either DOI or URL of the article must be cited.
License URL: https://creativecommons.org/licenses/by/4.0/

Keywords: Dysexecutive Luck hypothesis, Executive functioning, Event-related potentials (ERP), Anterior cingulate cortex (ACC), Stroop, Luck

Funding: National Council of Science and Technology The CONACYT in Mexico (National Council of Science and Technology) fully sponsors the postgraduate research degree under which this study is completed. The funders had no role in study design, data collection and analysis, decision to publish, or preparation of the manuscript.

==============================
There has been initial evidence to support the Dysexecutive Luck hypothesis, which proposes that beliefs in being unlucky are associated with deficits in executive functioning (Maltby et al., 2013). The present study tested the Dysexecutive Luck hypothesis by examining whether deficits in the early stage of top down attentional control led to an increase of neural activity in later stages of response related selection process among those who thought themselves to be unlucky. Individuals with these beliefs were compared to a control group using an Event-Related Potential (ERP) measure assessing underlying neural activity of semantic inhibition while completing a Stroop test. Results showed stronger main interference effects in the former group, via greater reaction times and a more negative distributed scalp late ERP component during incongruent trials in the time window of 450–780 ms post stimulus onset. Further, less efficient maintenance of task set among the former group was associated with greater late ERP response-related activation to compensate for the lack of top-down attentional control. These findings provide electrophysiological evidence to support the Dysexecutive Luck hypothesis.

Introduction

Estimations of the prevalence of beliefs about luck in the general population are as follows: 50% of people consider themselves lucky, 36% consider themselves neither lucky nor unlucky, and the remaining 14% consider themselves to be unlucky (Wiseman, Harris & Middleton, 1994). The most prominent theory within the psychological literature on beliefs around luck is irrational belief theory, which forms part of Rational Emotive Behavior Therapy (Ellis, 1994). According to this theory, beliefs around luck reflect absolute beliefs about the world, where many aspects of life are attributable to chance. Luck has an external, unpredictable, and uncontrollable influence upon the individual, eventually forming the basis of emotional distress (Ellis, 1994). A more recent distinction has been made between a perception of being lucky (belief in good luck) and the perception of being unlucky (belief in bad luck) (Darke & Freedman, 1997), with the latter being found to be associated with emotional distress, in terms of both hedonic and eudaimonic well-being (Maltby et al., 2008).

Consequently, there is literature that has focused on this distinction. One theory, the Dysexecutive Luck hypothesis (Maltby et al., 2013), focuses on belief in being unlucky. The Dysexecutive Luck hypothesis proposes that believing oneself unlucky is associated with lower levels of executive functioning. There are two possible causal directions underpinning the Dysexecutive Luck hypothesis. First, individuals’ deficits in executive functioning may have a negative influence on their ability to successfully achieve goals; as a result, they label themselves unlucky. Second, individuals believing themselves to be unlucky fail to engage the executive functions required for successful completion of key goals. Maltby et al. (2013) showed some initial support for the Dysexecutive Luck hypothesis. In that paper, self-report dysexecutive symptoms accounted for the unique variance in beliefs in being unlucky after controlling for established correlates of luck beliefs (personality, irrational beliefs, and self-efficacy). Experimental support for the Dysexecutive Luck hypothesis was demonstrated via significant positive correlations between individuals’ beliefs of their being unlucky, two (shifting and inhibition) of the three components (shifting, updating, and inhibition) of Miyake et al.’s (2000) taxonomy of executive functions, and a significant negative correlation with decision-making ability using somatic markers (Somatic Marker hypothesis; Damasio, Everitt & Bishop, 1996).

The Dysexecutive Luck hypothesis places a key emphasis on the role of executive functioning. Given the physiological basis of executive functioning (Jurado & Rosselli, 2007), the consideration of whether the Dysexecutive Luck hypothesis occurs at a physiological level is important, if only to elucidate the psychological nature of the hypothesis. Currently, there is vicarious physiological evidence for the hypothesis arrived at using measures (e.g., Switch cost task, Stroop test, and IOWA Gambling Task) that are well-established proxies for physiological functioning. For example, task-switching activates a common set of brain regions during diverse executive control operations, including medial prefrontal, superior and inferior parietal, medial parietal, and premotor cortices (Wager, Jonides & Reading, 2004). Functional magnetic resonance imaging (fMRI) studies have shown that the IOWA Gambling Task is related to aspects of the prefrontal cortex (Li et al., 2010) and the color-word Stroop test is related to activation in the frontal lobe of structures such as the dorsolateral prefrontal cortex (DLPFC) and the anterior cingulate cortex (ACC) (Spreen, Strauss & Sherman, 2006; Lansbergen, Van Hell & Kenemans, 2007; Silton et al., 2010).

To begin this consideration, we addressed the Dysexecutive Luck hypothesis in terms of attentional control, because Maltby et al. (2013) found evidence for such hypothesis around key attentional processes. The cascade-of-control model proposes that during tasks that demand attention, the DLPFC takes a leading role in implementing top-down attentional control and later ACC activity is thought to be involved in resolving response-related attentional processes (Banich, 2009; Milham & Banich, 2005; Silton et al., 2010). Previous studies assessing attentional control of the aging brain in healthy participants have provided evidence that during the color-word Stroop test, when early DLPFC activity is relatively low, late ACC activity increases (Milham et al., 2002; Silton et al., 2010). The authors suggest that increased late response conflict is a consequence of the reduced maintenance of a task set, which causes a need for increased ACC activity in order to compensate for the lack of top-down attentional control required to maintain adequate task performance (Milham et al., 2002; Silton et al., 2010).

The current study seeks to extend the Dysexecutive Luck hypothesis by providing initial electrophysiological evidence supporting it. For this goal, scalp recorded event-related potentials (ERPs) were recorded during the performance of a Stroop test. ERPs allow precise analysis of the time course of neural events since they provide a real-time temporal resolution of neural processes by reflecting event-locked electrical activity produced by neural ensembles (West & Alain, 1999). This electroencephalogram-derived technique was used thus to explore whether deficits in executive functions in ‘unlucky’ individuals are associated with elongated (slow) reaction time, and whether larger (more negative) late ERP waveforms underpinning response-related selection processes are specifically associated with the interference effect (in order to compensate for the lack of top-down control). The Stroop test was selected as the executive measure over other physiological assessments (i.e., task switch) since it is widely regarded as a prototypical inhibition task (a main function of attention control) (Miyake et al., 2000), a paradigmatic measure of selective attention (Carter, Mintun & Cohen, 1995), and a well-established proxy for physiological functioning. Its main effect (i.e., Stroop effect) is widely used in clinical practice and is the most extensively studied phenomena in experimental psychology (MacLeod, 1991). Most importantly, the Stroop test was utilized to test the Dysexecutive hypothesis for the first time in the study on which this study is based (Maltby et al., 2013) and aims to further develop.

Method

Participants

Initially, 217 undergraduate students (180 females; mean age = 20.34; SD = 2.1) completed the 6-item Beliefs in Being Unlucky subscale from the Beliefs Around Luck scale (Darke & Freedman, 1997; Maltby et al., 2008). Responses are scored on a 5-point scale (1 = “Strongly Disagree” to 5 = “Strongly Agree”). Internal reliability estimates for this subscale are α > .85 and the measure demonstrates validity via acceptable correlations between both peer and family ratings and predicted correlates of beliefs in being unlucky (Maltby et al., 2008). Participants were recruited from a university experiment participation scheme. The study was advertised and completed online via an electronic survey system.

Twenty-five students were selected from the initial group, all of whom had the highest (item mean >3.4) or lowest scores (item mean <1.8) on the Beliefs in Being Unlucky subscale. Five respondents’ data was discarded and not considered in the statistical analysis due to either excessive eye-movement artifacts or inconsistent scores in a re-test of their beliefs around luck. Thus, 20 students (18 women; mean age = 22.7; SD = 2.81) were selected for the final study. Two final experimental groups were formed: the 10 participants with the highest scores on the Beliefs in Being Unlucky subscale (‘unlucky’ group; mean = 4.63; SD = .37) and the 10 with the lowest scores on the Beliefs in Being Unlucky subscale (Control group; mean = 1.43; SD = .47). All participants had normal or corrected-to-normal vision, were unaware of the main purposes of the study, and had no history of any mental or neurological disorders. All participants, except one, were right-handed as assessed by the Edinburgh Handedness Inventory (Oldfield, 1971).

To establish further validity for the experimental groups we administered measures of dysexecutive symptoms (Wilson et al., 1996), neuroticism (Gosling, Rentfrow & Swann, 2003) and self-efficacy (Chen, Gully & Eden, 2001). This was to profile both experimental groups in terms of those characteristics found within the wider luck literature, namely that the belief in being unlucky is positively associated with neuroticism and dysexecutive symptoms, and negatively associated with self-efficacy (Maltby et al., 2008; Maltby et al., 2013). A series of Mann–Whitney U comparisons demonstrated that ‘unlucky’ individuals scored significantly higher on dysexecutive symptoms (U = 3.88 p < .001), neuroticism (U = 18.80, p = .016), and significantly lower on self-efficacy (U = 4.00, p < .001) than the control group.

Participants were paid 12.00 GBP for this study. The University of Leicester granted Ethical approval to carry out the study within its facilities (Ethical Application Ref: jm420-c5a3d).

Experimental measures

The aim of the experimental measures was to record event-related potentials (ERPs) during a manual color-word Stroop test. The construction of the current Stroop design was based on previous Stroop ERP investigations (West & Alain, 1999; Badzakova-Trajkov et al., 2009) but varied on two significant aspects.

The first modification was the addition of a neutral condition to the paradigm in order to fully match the stimuli conditions which now each respectively and separately holds a baseline comparison (two experimental conditions and two control conditions), allowing more flexibility and a more adequate statistical analysis. These two control conditions were required to be also balanced to the main conditions in perceptual difficulty/conflict, so that they correspondingly represent either one perceptive congruency or one perceptive incongruence only between them (See Table 1). Therefore, control incongruent trials provided one cognitive incompatibility: a grapheme-length mismatch; and control congruent trials delivered also a single cognitive compatibility: a grapheme-length match; whereas the incongruent and congruent conditions respectively possess either two incompatibilities (one grapheme-length and one semantic content mismatch) or two compatibilities (one grapheme-length and one semantic content match).

Table 1 Examples of stimuli conditions used during the experimental phase of the Stroop task.

Stimulus condition	Word stimuli examples	Description	
Congruent	‘red’ (in color red)	Word-color: matched Word-color grapheme length: matched	
Incongruent	‘red’ (in color blue)	Word-color: mismatched Word-color grapheme-length: mismatched	
Control congruent	‘dog’ (in color red), ‘jump’ in color (blue)	Word-color: neutral Word-color grapheme-length: matched	
Control incongruent	‘north’ (in color red), ‘deep’ (in color yellow)	Word-color: neutral Word-color grapheme-length: mismatched	

The second variance was the number of stimuli words matching for all conditions. The design had the same number of control non-color words as the color words in the experimental conditions (4) for a fully matched balanced design. Control non-color words were chosen carefully from the HAL online database; were all stimuli matched in similar HAL frequency, same number of syllabus and same part of speech (all stimuli were adjectives/nouns). Furthermore, two parallel versions of the Stroop were (still fully-matched) created differing only in the control non-color words, with the intention of avoiding any possible attentional bias due to a more distracting word.

Verbal stimuli for the Stroop test consisted of four color-words and four control non-color words. To reduce potential cognitive confound and bias among the verbal stimuli, the selection criterion for the control words was based and balanced on the following linguistic characteristics from the English Lexicon Project website (Balota et al., 2007): (1) HAL Word Frequency, i.e., selecting words with similar frequency values from the Hyperspace Analogue to Language frequency forms corpus (Lund & Burgess, 1996); (2) Parts of Speech, i.e., selecting only adjectives and nouns; (3) Lexical Decision Task Behavioural Results, i.e., similar mean RTs (ms); (4) Grapheme-length, words that are matched by the same number of letters to their respective color-word; and (5) Syllables, all of the control words matched the number of syllables of their respective color-word.

Procedure

Participants were instructed that it was highly important during the recording to remain still, avoiding as much body movement as possible, to keep their eyes fixated at the center of the screen, and to blink only when necessary and preferably between trials when stimuli disappeared.

Subjects sat 57 cm from a computer screen, and were restricted by a head and chin rest installed between the chair and the screen to minimize head movements. Stimuli were presented at a resolution of 1,024 × 768 pixels on a 21-inch monitor with a vertical refresh rate of 100 Hz. The task and EEG triggering was constructed and generated by E-Prime 2.0 software (Psychological Software Tools, Pittsburgh, Pennsylvania, USA), running on a PC Pentium IV desktop computer. Responses were recorded using a serial response box that featured a 0 ms debounce period, which allows for a high precision in answer recording. Each trial started with 1,000 ms of a white fixation cross (“+”) which was presented over a black background at the center of the screen. This was replaced with the target stimulus item shown for 1,000 ms. A blank screen followed lasting 700 ms. Written feedback with a duration of 300 ms appeared at the center of the screen after incorrect (‘wrong’) and missing (‘missing’) responses. The speed of response timeframe remained unaltered for all trials with a total duration for each trial being 3,000 ms.

Participants were required to indicate the color of the font (red, blue, yellow, or green) of the stimuli shown at the center of the screen. They did so by pressing one of the four corresponding colored buttons on a button box using the index and middle fingers of the right and left hands. All target stimuli were presented in a random order, although all were presented in the font Courier New, with a font size of 25 and a bold font style. At the beginning of the acquisition and practice phases, and before the start of each test block, a message appeared on the screen instructing the participants to press the space bar to begin the block of trials. After the space bar was pressed, the screen was blank for 2,000 ms.

The total duration of each Stroop experimental session recording was 30 min, and consisted of three sequential phases:

(a) A color-to-key acquisition phase designed to establish a strong mapping between the stimulus colors and the correct response keys. This phase was presented in a single block of 96 trials with each of the four colors represented 24 times in a series of ‘X’s, equal in grapheme-length to the color’s name (‘xxx’ in red font, ‘xxxx’ in blue font, ‘xxxxx’ in green font, and ‘xxxxxx’ in yellow font).

(b) A practice phase consisting of 48 trials with the four types of stimuli conditions (congruent, incongruent, control congruent, and control incongruent) also used in the upcoming phase. Stimuli in the congruent condition were colored words that were presented in the color congruent to their meaning (e.g., ‘RED’ in red font). Stimuli in the incongruent condition were colored words presented in any of the three colors that did not match their meaning (e.g., ‘RED’ in blue font). Stimuli in the control congruent condition were colored words that were matched for their grapheme length with the color words (e.g., ‘DOG’ in red font). Finally, stimuli in the control incongruent condition were colored words that were mismatched for grapheme length with the color words (e.g., ‘NORTH’ in red font). Each of these four conditions had 12 trials.

(c) A test phase (EEG recorded) that had the 4 condition trials balanced, with each presented 24 times in a single block, for a total of 96 trials per block. A brief break occurred in between blocks. Six blocks with a grand total of 576 trials were run in this phase. Stimuli in all phases were presented in a random order.

ERP recording and analysis

Continuous EEG signals were recorded by a DC 32-channel amplifier (1-kHz sampling rate, 250 Hz high cut-off frequency; Brain Products Inc., Gilching, Germany). The EEG activity was recorded via a Waveguard elastic cap, containing 64 unshielded and sintered Ag–AgCl electrodes (CAP-ANTWG64; ANT, Netherlands), with an electrode layout according to the international 10–5 electrode system. The following electrodes were used during the recording FP2, F3, FZ, F4, FC5, FC1, FCZ, FC2, FC6, C3, CZ, C4, CP5, CP1, CP2, CP6, P7, P3, PZ, P4, P8, PO7, PO3, PO4, PO8, O1, OZ, and O2. The right-earlobe electrode served as on-line reference. EEG waveforms were re-referenced off-line to the average of the right- and the left-earlobe electrodes. Two electrodes placed in a bipolar montage at approximately 1 cm from the outer canthus of each eye served to record the horizontal electrooculogram (HEOG). The vertical electrooculogram (VEOG) and blinks were recorded from one electrode positioned below the right eye and Fp2 referenced to the right earlobe. Electrode impedance was kept below 5 KΩ. EEGs were epoched from 200 ms pre-stimulus-onset to 1,000 ms post-stimulus-onset. Each EEG epoch was inspected off-line, and those with ocular artifacts (as indicated by HEOG activity exceeding ±30 µV and VEOG activity exceeding ±80 µV) were excluded from statistical analyses. Only ERP data for trials with correct responses were analyzed, therefore artifacts from eye movements and excessive noise were marked bad and discarded after a selective individual trial review of each participant’s data. To help remove slow and sustained shifts in voltage (from non-neural origin) during data acquisition and reduce high-frequency noise, averaged ERPs were filtered using 0.05 Hz high-pass, 30 Hz low-pass and 50 Hz notch filters.

Mean amplitudes of ERP waveforms in the time window of 450–780 ms relative to a 200 ms pre-stimulus baseline were obtained for each subject under each of the four conditions. This late ERP time window was chosen based on visual inspection and relevant source-ERP color-word Stroop research. In particular, a recent study (Silton et al., 2010) supported a role only for late ACC activity (520–680 ms), which is related to later aspects of response selection, in differentiating Stroop conditions. ERP mean amplitudes were measured for a selected group of 4 electrodes in the fronto-central scalp region (Fz, FC1, FCz, FC2). This region was chosen because it was likely to reveal the brain processing associated with cognitive control in a situation requiring effective inhibition of distracting task-irrelevant information (Badzakova-Trajkov et al., 2009).

Variable creation and statistical analyses

A semiautomatic filtering operation of raw reaction times (RTs) data was carried out by E-prime 2.0 in order to remove extremely slow (retardations) and extremely fast (anticipations) responses. Consistent with other studies in the area (e.g., Fuggetta, 2006), an absolute exclusion criterion excluded RTs less than 150 ms and greater than 3,000 ms. The magnitudes of a variety of effects were computed and analyzed for analysis. These effects were obtained after computing three different combinations of conditions and mean averages in the Stroop test, such as the facilitation effect (control congruent condition minus congruent condition), interference effect (incongruent condition minus control incongruent condition), and Stroop effect (incongruent condition minus congruent condition). Mean error (%), and Stroop condition effects (ms) for the correct trials were only used in the reported analysis of variance (ANOVA). Three differential effects of the factor condition on the two groups of participants were quantified on the basis of ERP mean amplitudes from a particular region of interest (fronto-central area) using four electrodes’ activity (Fz, FC1, FCz, and FC2) during the time window of 450–780 ms post-stimulus-onset and reading that were reported in the ANOVA. For each ANOVA, the sphericity assumption was assessed using Mauchly’s test. Greennhouse-Geisser epsilon adjustments for non-sphericity were applied where appropriate. Post hoc paired t-test were Bonferroni-corrected for multiple comparisons. For statistical testing, p < .05 was considered significant.

Results

The mean percentage of errors in the Stroop test was 2.572 ± 0.574% with the mean percentage of errors for the facilitation effect M = 0.31 (SD = .62), interference effect M = 3.17 (SD = 0.8) and Stroop effect M = 4.24 (SD = 1.5). An ANOVA showed a significant effect for the conditions (F(1.2,21.6) = 7.24, p = .002, ŋp2 = .29), with post hoc comparisons showing a significant difference between facilitation effect compared to the interference effect (.308% vs. 3.167%, p = .033), and the Stroop effect (.308% vs. 4.242%, p = .037). This means that participants in both groups had significantly more errors in the interference and Stroop conditions than in the facilitation condition (Stroop and interference conditions represent the conflict). These effects are expected for each Stroop test. There was no significant main effect for group (F(1,18) = 1.24, p = .280), meaning both groups performed in a similar fashion across the conditions.

In terms of the experimental groups’ performance on the Stroop test, descriptive statistics (mean, accuracy, and RTses) for the three Stroop condition effects in the two groups are shown in Table 2. Mean RTs for the magnitudes of the three condition effects on the Stroop test for both groups were 78.15 ± 9.39 ms. An ANOVA for the mean RTs showed a main effect of ‘Group’ (F(1,18) = 6.30, p = .022, ŋp2 = .26). There was also a significant main factor “condition effect” (F(1.2,21.2) = 55.9, p < .001, ŋp2 = .76). Furthermore, there was a two-way interaction “condition effect by group” with F(1.2,21.2) = 5.3, p = .027 ŋp2 = .23. Post hoc comparisons showed a significant difference between the ‘unlucky’ and control group for interference effect (68 vs. 132 ms, p = .027), and Stroop effect (90 vs. 163 ms, p = .017), but not in the facilitation effect (6 vs. 9 ms, p = .766).

Table 2 Mean reaction times in milliseconds and mean accuracy (%) for both experimental groups.

	Main condition effects	
	Interference (incongruent condition— control incongruent condition)	Facilitation (control congruent condition—congruent condition)	Stroop (incongruent condition— congruent condition)	
Control group	
Reaction time (ms) (SD)	67.77 (48.78)	6.00 (16.80)	90.01 (56.87)	
Errors % (SD)	6.10 (1.13)	−1.33 (0.88)	5.5 (1.48)	
‘Unlucky’ group	
Reaction times (ms)	132.47 (69.90)	9.14 (28.40)	163.50 (67.66)	
Errors %	3.92 (4.55)	.77 (2.27)	4.70 (5.36)	

Figure 1 shows the grand average ERPs of the three main condition effects for both experimental groups. The time window where the greatest ERP activity occurred was the late time window, 450–780 ms, which showed a statistically significant main effect of ‘group,’ with F(1,18) = 7.17, p = .022, η2 = .285. There was an overall effects significant difference with decreased ERP amplitudes for the control group compared with the ‘unlucky’ group (0.169 vs. −2.674 µV). More importantly, a significant three-way interaction ‘Electrode * Condition effect * Group’ was also found (F(6,108) = 2.21, p = .028 η2 = .109). Post hoc comparisons of the significant three-way interaction revealed a significant difference in the amplitude of the late ERPs between the ‘unlucky’ group and the control group. Notably, there was a significant difference in the magnitude of interference effect for electrodes Fz (0.202 vs. −3.788 µV), FC1 (−0.335 vs. −3.638 µV), FCZ (−0.078 vs. −3.663 µV), and FC2 (−0.324 vs. −3.752 µV), respectively (the study’s region of interest). Moreover, there was a significant difference in the magnitude of the facilitation effect between individuals with no beliefs in being unlucky and individuals with beliefs in being unlucky for electrode Fz (0.608 vs. −2.951 µV) and FC1 (0.753 vs. −2.821 µV).

Figure 1 Image of grand average ERPs.

Grand average erps of the three main condition effects for both study groups. Notes: All the effects were obtained from the combination of four fronto-central electrodes, or region of interest (FZ, FC1, FCZ, and FC2) during the time window of 450–780 ms. the greatest effect was found on the interference effect (A). The next figure (see Fig. 2) shows individual erp’s of each of these four main electrodes.

Figure 2 Image of individual electrodes ERPs.

Four main individual electrodes’ erp’s of each stroop test condition (A1, A2, A3, and A4) and of the three main condition effects (B1, B2, B3, and B4) for both study groups.

Discussion

The first finding, consistent with Maltby et al. (2013), was that ‘unlucky’ individuals performed poorly on the Stroop test and experienced higher interference main effects when compared to the control group. The interference effect differences between groups are of great importance since it has been emphasized that the overall Stroop effect is not enough for an accurate conflict measure; it is necessary to have a comparison to a baseline neutral condition. The interference effect, which measures an interference or “cost” relative to a neutral condition, is the most reliable and robust component within the Stroop test (Henik & Salo, 2004; MacLeod, 1991). Electrophysiological results supported the role of late negative ERPs’ amplitude due to incongruent trials in differentiating ‘unlucky’ individuals from the control group in the magnitude of the interference effect.

In the current study, as in previous ERP studies, we used a difference waveform (incongruent condition—control incongruent condition waveforms) in an attempt to isolate the processes specifically associated with the interference effect (Badzakova-Trajkov et al., 2009; Markela-Lerenc et al., 2004). Although the late time window chosen in the current investigation (between 450 and 780 ms) is an unusual epoch for Stroop-related effects, we were particularly interested in assessing late stage response-related processing, which we hypothesized would be enhanced in the group of ‘unlucky’ individuals to compensate for their impairment in high-order attentional control processes. A recent study employed the source-waveform ERP mediation analysis (Hayes, 2013) and found that only late ACC activity (520–680 ms) was correlated with the interference effect, distinguishing the waveforms between incongruent/congruent conditions (Silton et al., 2010). Thus the findings demonstrated, as did previous studies, that late stage response selection processes are specifically associated with ACC function (Milham et al., 2003; Silton et al., 2010). Furthermore, an ERP study on the color-word Stroop test (Liotti et al., 2000) found a significant difference in the incongruent relative to the congruent trials on a left temporoparietal cortex scalp region during a late time window of 600–700 ms, supporting the role of late stage response selection processes in the interference effect.

Overall, both the indirect behavioral and direct electrophysiological significant interference effects of the present study can be interpreted with the Silton et al. study (2010), which integrated fMRI and ERP data to identify the time course of regional brain activity associated with top-down attentional control during the execution of a color-word Stroop test. Silton et al. demonstrated that the degree to which ACC influenced Stroop performance depended on the level of earlier DLPFC activity. When DLPFC activity levels were high, there was little ACC impact on Stroop performance. This suggests that when DLPFC provides sufficient attentional control, ACC has a smaller effect on overt performance. The finding that ACC activity was not critical for performance when DLPFC activity was high is also consistent with a prior study (Milham et al., 2003). A similar pattern of neurophysiological activity has been found in the current study in the individuals of the control group. This group’s minimal level of late ERP interference effect and adequate level of behavioral performance are direct evidence for this interpretation.

Furthermore, Silton et al. (2010) demonstrated that in the case of low DLPFC activity, there was a relatively high late ACC activity, which affected Stroop performance with a response pattern that involved slow RT responses. These results were consistent with the idea that ACC was compensating for the lack of top-down DLPFC control (Silton et al., 2010). The pattern of results for ‘unlucky’ individuals in the current ERP study echoes these previous results; the individuals exhibited a significantly increased response conflict with greater RTs and larger late negative response-related ERPs during incongruent trials, compared to the control group.

The increased facilitation effect exhibited only in ERPs could be the result of an early, nonstrategic priming effect (word ‘red’ in red font), or a deficiency in the strategic allocation of attention that may result in word-reading errors (participants read the word rather than name the color). In terms of the interference effect, both the RTs and ERP results complement each other quite well, given that it is the only one of the three main condition effects revealing significant differences between groups. A global interpretation of these results suggests that ‘unlucky’ individuals had a slower processing of the interference or “cost” relative to a baseline condition in the interference effect, translating to slower RTs and greater ERP magnitudes in this condition effect when compared to the control group. Furthermore, the control group had adequate attentional resources available to perform the task and quickly resolved the ‘conflict condition,’ and did not manifest a strong interference effect in the late ERPs. This suggests that the conflict resolutions of the Stroop test occurred earlier in the control group than in the group of ‘unlucky’ individuals.

The present study is observational and cross-sectional in nature (since no conditions were experimentally manipulated), therefore it is important to highlight that the findings cannot establish that neural dysfunction in the executive network caused participants beliefs in being unlucky. Neither confirm, at a highly specific localization level, which specific structures of the brain were responsible for the different main effects found in this investigation (mainly due the limited number of channels (24) used in the electrophysiological data recording). However, these effects do show an association between self-reported beliefs and neural dysfunction, and also suggest an anatomical basis. The electrodes’ region of interest in the current findings (fronto-central scalp) is in agreement with other ERP investigations that reveal a consistent neuroanatomical basis, correlating the Stroop effect with strong activation in the DLPFC and especially the ACC (e.g., Carter & Van Veen, 2007; Botvinick et al., 1999). Taken together, these studies strongly support a conflict-monitoring hypothesis (Bush, Luu & Posner, 2000), whose main premise identifies the ACC as the structure of the brain responsible for signalling the occurrence of conflicts in information processing, triggering compensatory adjustments in cognitive control. In a Stroop test, this event can be specifically identified in the semantic conflict generated by the incongruent stimuli.

With respect to future research related to the current thesis work, there is an opportunity to further examine some of the findings presented. For instance, a possible study based on two pending ideas would consist on first seeing if the electrophysiological lead supporting the dysexecutive luck with belief in bad luck in UK can be replicated in another country, and/or the potential to extend the number of luck factors employed in that study and work with all four beliefs around luck factors. This would signal if there were cross-cultural potential to the earlier work and something that could be pursued, as a continuation of this thesis work. Additionally, it would be also interesting to examine how the dysexecutive hypothesis might behave when applied to other related and common beliefs, such as hope.

In summary, the findings provide physiological data that supports the dysexecutive luck hypothesis. They suggest that increased response conflict in the context of deficits executive functions of ’unlucky’ individuals have probably caused a need for increased late ACC activity, which in turn translated to lengthened RTs and increased magnitude of late ERPs primarily involved in response-related processes. This alteration of regional neural activity also supports the concept introduced in previous studies that ACC was compensating for the lack of DLPFC attentional control in the attempt to maintain adequate task performance.

Supplemental Information

Supplemental Information 1 Belief in Luck Scale Screening

Click here for additional data file.

Supplemental Information 2 Scales and Stroop Data containing Stroop RTs and Stroop’s main conditions.

Click here for additional data file.

Supplemental Information 3 Stroop EEG microvolt data for all the Stroop main condition effects

Click here for additional data file.

Additional Information and Declarations

Competing Interests

Author Contributions

Human Ethics

The authors declare there are no competing interests.

Jaime Martín del Campo Ríos conceived and designed the experiments, performed the experiments, analyzed the data, contributed reagents/materials/analysis tools, wrote the paper, prepared figures and/or tables, reviewed drafts of the paper.

Giorgio Fuggetta conceived and designed the experiments, performed the experiments, analyzed the data, contributed reagents/materials/analysis tools, prepared figures and/or tables, reviewed drafts of the paper.

John Maltby conceived and designed the experiments, contributed reagents/materials/analysis tools, wrote the paper, reviewed drafts of the paper.

The following information was supplied relating to ethical approvals (i.e., approving body and any reference numbers):

The University of Leicester granted Ethical approval to carry out the study within its facilities (Ethical Application Ref: jm420-c5a3d).

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
