# Peer review of "Beliefs in being unlucky and deficits in executive functioning: an ERP study"

_PeerJ, doi:10.7717/peerj.1007_

## Round 0.1 · original submission · Major Revisions

Please refer to the comments from the two reviewers and revise the manuscript accordingly.

·

Basic reporting

This paper meets basic reporting standards required by PeerJ, however the English needs some serious editing for clarity. The main body of the manuscript is centered rather than left aligned or full justified. This should be corrected.

The introduction allows the reader to understand the experimental design. However, I do wonder why the Stroop was chosen, instead of a task switch. Do the authors plan to test this hypothesis in future with other executive measures?

The background on ERP association with Stroop is sparse, and could be more fully developed. Is there any literature on ERP association with the Dysexecutive Luck hypothesis?

Experimental design

The question of whether beliefs are associated with neural function is important. Indeed, cognitive neuroscientists are tasked with discovering how neural function and cognitive behaviors such as belief formation and activation are related. This research question addresses how beliefs about being unlucky may be rooted in neural dysfunction in brain microcircuit arrays shown to subserve executive function. The investigators selected a Stroop paradigm as an executive function exemplar. Stroop evaluates inhibitory function. Others might have been selected, such as a task switch, which can evaluate the interaction between inhibitory, updating, and shifting functions. The authors may want to make it particularly clear why they selected the Stroop over other executive assessments and what future research directions they plan to take.

For their experiment, self-reported beliefs of being lucky or unlucky were the basis for dividing participants into two groups: self-reported lucky or unlucky. Participants were then administered a carefully quantified Stroop test (facilitation, interference, and Stroop effects) during EEG recording. Their parsing of the Stroop is excellent. Selection of the EEG event-related time window is reasonable given their expected neural mechanisms.

This study thus appears to be observational and cross-sectional in nature. No conditions were experimentally manipulated.

Validity of the findings

Because this study is not a true experiment, it cannot establish that neural dysfunction in the executive network caused participant beliefs in being unlucky. All they can do is show that there is an association between self-reported beliefs and neural function.The authors need to be very careful to make this clear in their discussion. Designing an experiment to establish neural causation of any belief will be difficult in humans.

That being said, the investigators do show a very clear difference between their groups in the direction they predicted on neural function as well as behavioral performance.

Additional comments

I suggest you find a very good editor to copy-edit your manuscript for clarity.

I suggest you report correlations between your Stroop measures, ERP amplitudes, and self-reported luck or unluck belief.

Reviewer 2 ·

Basic reporting

This study tested the Dysexecutive Luck hypothesis using by ERP. It investigates individuals who thought themselves to be unlucky if deficits in the early stage of top down attentional control led to an increase of neural activity in later stages of response related selection process. This article does meet the standards.

Experimental design

The study performed A Stroop test and was methodologically well elaborated. However, in line 156-176: Regarding to the Stroop experimental, consisting of three sequential phases, the author did not explain the reason for such design and state their consideration.

Validity of the findings

1. Line 271-273: “Electrophysiological results supported the role of ………..of Stroop interference effect.” Is the term ‘Stroop interference effect’ the same as “interference effect”? Does it different from the term ‘Stroop effect’ in line 216?
2. Figures A- C showed the Grand Average ERPs of the three condition effects for both groups. However, these effects were obtained after computing three different combinations of conditions and mean averages in the Stroop test, such as the facilitation effect (control congruent condition minus congruent condition), interference effect (incongruent condition minus control incongruent condition), and Stroop effect (incongruent condition minus congruent condition) across behavioral and ERP data respectively. The weakness of this approach in ERP waveform analysis, however, is that physiological processes are usually not additive, that is, do not occur such that the physiological processes in one condition equal those processes in the other conditions plus or minus one other process. Consequently, the authors should provide some evidence to support the straightforward interpretation of the difference waveform. Moreover, if subtractions are used, the original ERPs from which the difference waveforms were derived should be presented together with the difference waveforms. Four electrodes’ original ERPs (Fz, FC1, FCz, and FC2) during the time window of 450-780 ms post stimulus onset on four conditions must be shown.
3. With regard to RT, a significant difference between the 'unlucky' and control group for interference effect and Stroop effect, but not in the facilitation effect. However, there was a significant difference in the magnitude of interference effect for electrode Fz, FC1 , FCZ, and FC2. Also, there was a significant difference in the magnitude of the facilitation effect between individuals with no beliefs in being unlucky and individuals with beliefs in being unlucky for electrode Fz and FC1. But no significant difference in the magnitude between the 'unlucky' and control group for Stroop effect. Results showed discordant findings between RT and ERP in both Stroop effect and facilitation effect. What’s the interpretation for these findings?

Additional comments

Results and discussion provide a few new insights. Therefore, the submitted paper is suitable for publication. However, there are a number of important issues and recommendations mentioned above that would seem to require careful attention for the final version.

---

## Round 0.2 · accepted · Accept

Congratulations and thank you for your submission.

Reviewer 2 ·

Basic reporting

The authors addressed several points raised properly, and the manuscript is now much clearer.

Experimental design

The revision has profited from the reviewers comment, and demonstrating stronger main interference effects in the unlucky group with more negative distributed late ERP component during incongruent trials represents strength of the present paper.

Validity of the findings

I recommend the revised paper could be published.